

# Mineral, thermal and deep groundwater of Hesse, Germany

Rafael Schäffer[1], Kristian Bär[1], Sebastian Fischer[2], Johann-Gerhard Fritsche[3], Ingo Sass[1]

[1]Geothermal Science and Technology, Technical University of Darmstadt, Darmstadt, 64287, Germany
[2]Federal Institute for Geosciences and Natural Resources, Berlin, 13593, Germany
[3]Hessian Agency for Nature Conservation, Environment and Geology, Wiesbaden, 65203, Germany

*Correspondence to*: Rafael Schäffer (rafael.schaeffer@tu-darmstadt.de)

**Abstract.** The composition of mineral, thermal or deep groundwaters is of interest for several geotechnical applications, such
as drinking water supply, spas or geothermal energy. A verified and reliable knowledge of temperature, pH, hydrochemical
composition and other parameters is crucial to extract fluids with as few technical problems as possible and exploit
groundwater reservoirs economically and environmentally sustainable. However, at sites where empirical data are lacking, the
correct prediction of fluid properties is often difficult, resulting in considerable economic risks. Here we present the first
comprehensive and publicly available database of mineral, thermal and deep groundwaters of Hesse compiled from published
and own data. Presently, it contains 1035 datasets from 560 different springs or wells sampled since 1810. A dataset consists
of metadata like location, altitude, depth, rock type or stratigraphic unit, information on the water type, references, physical-
chemical parameters, concentrations of major, minor and trace elements, content of dissolved and free gases as well as isotope
data. The dataset allows the evaluation of time series and distribution of groundwater properties both laterally and vertically.
We show a simple statistical evaluation based on the five major hydrogeological regions of Hesse. Our database can be used
to re-evaluate genesis and circulation of deep groundwaters, to estimate reservoir temperatures with a solution
geothermometer, or to assess groundwater ages by means of isotope data. It can also be useful for a first conception of deep
geothermal utilizations. In future, an update and extension of the database is intended.

**Short summary.** Knowledge of groundwater properties are relevant e.g. for drinking water supply, spas or geothermal energy.
We compiled groundwater data almost exclusively from publications resulting in 1035 datasets from 560 springs or wells
sampled since 1810. The database is useful to describe spatial or temporal variation of groundwater composition and to reduce
uncertainties. It also serves to re-evaluate the movement of deep groundwaters or the estimate its characteristics such as
temperature and age.




## 1 Introduction

### 1.1 Motivation

The groundwater characteristics is determining its suitability for drinking water, thermal water, mineral water, mineral extraction, heat extraction and so on. It also impacts largely its technical utilization since changes of temperature and pressure influence processes like scaling, corrosion or degassing, resulting in further costs for remediation measures. In addition, the hydrochemical composition also controls fluid properties like density, heat capacity or viscosity. Knowledge of the hydrochemistry is therefore imperative in order to foresee technical or economic challenges which might emerge during utilization. For this purpose, we shaped the database of mineral, thermal and deep groundwaters of Hesse, Germany. It compiles 1035 published hydrochemical data sets from 560 different measurement points in the entire Hessian territory and some adjacent areas. It has been compiled by the TU Darmstadt in close cooperation with the Hessian Agency for Nature Conservation, Environment and Geology (HLNUG) as part of the R&D project Hessen 3D 2.0 ('3D modelling of the petrothermal and medium deep geothermal resources for power production, direct heat utilization and storage of the federal state of Hesse') funded by the Federal Ministry for Economic Affairs and Energy (grant number 0325944A).

The data has been collected in close cooperation with the Federal Institute for Geosciences and Natural Resources (BGR). This cooperation emerged because BGR is setting up a comprehensive database containing quality-checked data about formation waters from the entire German territory (Fischer and May, 2019). BGR currently reviews internal resources (databases, library and archive) and plans to collect data from various external sources including Local State Authorities, research institutes/universities and industry. Within the formation water database formation water is defined as groundwater with TDS > 1 g l$^{-1}$ and/or temperature > 20 °C that originates from hydrogeologic entities deeper than 100 m bsl. The formation water database is being implemented within the GeODin software package and will be made publicly available in the future.

### 1.2 Geology and hydrogeology of Hesse

Fritsche et al. (2003) and Ad-hoc-Arbeitsgruppe Hydrogeologie (2016) divide Hesse into five major hydrogeological regions (HYR, see figure 1):

1) the Upper Rhine Graben including the Mainz Basin in southern Hesse as well as Cenozoic deposits in northern Hesse with the volcanic rocks of the Vogelsberg (HYR-03),
2) parts of the Central German Basin Range in eastern and northern Hesse, with dominantly Triassic rocks at the surface (HYR-05),
3) parts of the Southern German Cuesta Landscape in south-eastern Hesse (HYR-06),
4) the Rhenish Massif in western and north-western Hesse as part of the Variscan Rhenohercynian zone (HYR-08), and



5) the crystalline Odenwald and Spessart in south-eastern Hesse as part of the Variscan Mid-German Crystalline Rise (HYR-10).

**1.2.1 Upper Rhine Graben, Mainz Basin, Hessian Depression and Vogelsberg (HYR-03)**

The Quaternary unconsolidated sediments of the Upper Rhine Graben constitute a thick and productive, intensively used pore aquifer. Especially at the eastern margin of the graben, clayey intercalations divide the sequence of sand and gravel into separated aquifers with different pressure levels and varying water quality. In general, mineralisation increases with depth and some of these highly mineralised waters ascend along permeable fault zones active in the Cenozoic sediments. Groundwater-bearing Tertiary sands and gravels follow underneath, which often alternate with clay-rich layers and can therefore locally form natural gas deposits. Marls, mudstones and limestones of the deeper Tertiary strata feature little but highly mineralized groundwater. Layers of the Pechelbronn-Group are oil bearing. The Upper Rhine Graben is well explored by deep drillings due to the hydrocarbon exploration and exploitation since the 1950s. However, the Permian Rotliegend, which forms the base of the Cenozoic Graben deposits at approximately 2 to 3.2 km depth, has been drilled by only up to 70 wells so far but has rarely been sampled for its hydrochemistry. Even the most recent well, drilled to a depth of about 4,000 m deep geothermal energy production in Trebur in 2016 (Reinecker et al., 2019), has not been used for fluid sampling.

Limestone beds and gravel layers within Miocene and Oligocene marls feature fossil groundwater in the northern foothills of the Mainz Basin on the southern edge of the Taunus. To the south, the Taunus Border Fault separates the Rhenish Massif from the Upper Rhine Graben and locally acts as a vertical pathway for the rise of highly mineralized deep waters (Loges et al., 2012), reaching the highest temperatures in the west, e.g. Kochbrunnen in Wiesbaden with more than 65 °C (Wagner et al., 2005). The temperatures of mineral waters are at least 30 °C lower in eastern direction: Bad Soden (max. 32.0 °C), Bad Homburg (max. 22.4 °C) and Bad Nauheim (max. 32.7 °C) (Golwer, 2005, Schäffer and Sass, 2016).

Southeast of Frankfurt to Aschaffenburg, the Hanau-Seligenstadt Depression, filled by Pleistocene gravels and sands of the Main River and Tertiary unconsolidated rocks, form an up to 180 m thick sequence of pore aquifers, divided by clayey aquicludes above the Rotliegend. Similar sequences, formed by limnic-fluviatile sands and gravels as well as clays of the Tertiary exist in the Wetterau, situated in the transition area to the Hessian Depression. The thickness and depth of aquifers and aquicludes can vary greatly as a result of the local block faulting. In deeper Tertiary aquifers, highly mineralized waters with elevated NaCl concentrations occur, rising particularly in the western Wetterau along the eastern, partly covered, extension of the southern Taunus Border Fault (Scharpff et al., 1974). The important mineral water deposits of Bad Vilbel, Rosbach vor der Höhe and Bad Nauheim are located along this fault zone and conclude the line of thermal spas associated to this major fault (Chelius, 1905, Scharpff, 1976, Golwer, 2009).

In central Hesse, the thick sequence of Tertiary volcanic rocks of the Vogelsberg (e.g. basalts, interbedded pyroclastic materials and their weathering products) and the tectonic overprinting of the rock sequence cause a fractured, multi-layered aquifer system (Ehrenberg et al., 1981). Hydrogeologically, the Vogelsberg is subdivided into the zone of suspended aquifers and the zone of continuous groundwater saturation (Leßmann et al., 2001). The Vogelsberg is considered as a supra-regional water



extraction area and thus of great significance for water management. The volcanic rocks are underlain by Triassic Buntsandstein. The 1,000 m deep thermal water well in Herbstein taps mineral water from Zechstein and Rotliegend with a temperature of up to 33 °C (Hölting, 1979, Käß and Käß, 2008).

The Hessian Depression is a Cenozoic Graben structure filled with thick deposits of Tertiary sediments, which cover the

palaeosurface of the Buntsandstein. Usually, the Tertiary strata are rather unproductive aquifers. Poorly permeable rocks of the Upper Buntsandstein (Röt) with thicknesses between 100 and 200 m form the base of this aquifer. Below follows the Middle Buntsandstein aquifer, which is used for drinking water production. Groundwater recharge mainly occurs through inflow from catchment areas in the west and east outside the Hessian Depression. Tertiary basalts, which cut along fault lines through the sedimentary sequence, form fractured aquifers of low extent and yield. Two circa 1,300 m deep boreholes tapped

Na-Cl waters in dolomites of the Permian Zechstein with temperatures of nearly 42 °C in Kassel-Wilhelmshöhe at the beginning of the 20th century (Udluft, 1969).

### 1.2.2 Central German Basin Range (HYR-05)

The Central German Basin Range includes several hydrogeological subareas. Two groundwater levels are distinguished in the

Borgentreich Depression and Kassel Graben in northern Hesse. One aquifer exists in the Triassic Muschelkalk, another in the Middle Buntsandstein, and are largely separated from each other by the aquiclude of the Upper Buntsandstein. The north-south oriented Leine Graben in the most north-eastern part of Hesse consists of a mosaic of faulted blocks composed by Triassic sediments, largely undisturbed internally. In the Werra-Greywacke Hills between Witzenhausen and Eschwege, shallow groundwater from Palaeozoic rocks already has a high hardness and sulphate concentration due to overlaying carbonate rocks

of the Permian Zechstein. The same accounts for groundwater from siliceous fracture aquifers of the Rotliegend within the Richelsdorfer Gebirge.

The Lower and Middle Buntsandstein dominantly crops out in large parts of north-eastern and eastern Hesse. In this hydrogeological subarea known as the Fulda-Werra-Uplands, hydraulically separated groundwater levels are formed locally in individual faulted blocks within the Buntsandstein aquifer. However, seen on a large scale, these blocks are hydraulically

connected. Several roughly northeast-southwest and perpendicular oriented grabens with fault displacements of several hundred meters interrupt the faulted blocks of the Buntsandstein. In the Rhön Mountains, younger Triassic sediments and Tertiary basaltic rocks partly cover the Buntsandstein. The Zechstein contains salt deposits in stratiform bedding in depths between 200 and 1,000 m between Bad Hersfeld and the Thüringer Wald (Thuringian Forest) as well as south of the Fulda-Pilgerzell Graben near Neuhof (Trusheim, 1964; Käding, 2001). The fault tectonics are less pronounced in this area due to the

plastic behaviour of the salt deposits. This causes a lower hydraulic permeability of the overlying Triassic strata. Over a large area, clayey sequences within the Zechstein protect these salt deposits from dissolution. Leaching leads to a disintegration of the caprocks only at the edges of the salt slope (the inclined rim of the salt deposit exposed to leaching) or in areas of irregular subrosion depressions, increasing the rise of deep saline groundwater from dolomites of the Leine-Formation (Plattendolomit). The Plattendolomit is a well-permeable fracture/karst aquifer that bears confined, partly artesian and highly mineralized





groundwater of the Na-Cl type (Hölting, 1981, Schäffer et al., 2018). Since the 1920s, it has also served as a horizon for saline wastewater injection from the potash industry in the Werra and Fulda area (Skowronek et al., 1999). Higher mineralized groundwater with saline wastewater components rises from the Plattendolomit and the Lower Buntsandstein to the surface particularly on the salt slope and at shallow depths of the Plattendolomit in river valleys (Skowronek et al., 1999).

### 1.2.3 South German Cuesta Landscape (HYR-06)

In south-eastern Hesse, clastic rocks of the Buntsandstein form a fractured multilevel-aquifer divided by clayey intercalations which is used for drinking water production (Ludwig et al., 2011). In the eastern Odenwald, the Buntsandstein overlies the Zechstein and the basement with increasing thickness towards the east. Muschelkalk is preserved within the Michelstadt

Graben and forms a shallow karst aquifer (Becht et al., 2017).

### 1.2.4 Rhenish Massif (HYR-08)

The Rhenish Massif forms northwestern Hesse. It consists of strongly folded, imbricated, and widely over-thrusted low-grade metamorphic rock sequences which are often dipping steeply and faulted narrowly. The Carboniferous slates in the northern

Rhenish Massif have a low hydraulic permeability. Of higher permeability are Devonian slates, silicified rocks and especially greywackes and limestones. The latter can be karstified to the point of forming caves. Zechstein sediments crop out at the eastern rim of the Rhenish Massif, northwards and eastwards of the Kellerwald. The karstified carbonate and anhydrite rocks, which also comprise clastic sediments, form fracture and karst aquifers. South of the Kellerwald dominates the marginal Zechstein facies consisting of siltstones and mudstones. Fracture aquifers of the Buntsandstein dominate further towards the

Hessian Depression, containing medium- to coarse-grained sandstones with a few thin layers of mudstone.

Over a large area, the rock sequences in the southern Rhenish Massif are producing even less groundwater and are less permeable than in the north. Only the well-fractured Taunus quartzite, constituting the ridge of the Taunus Mountains is used here for large scale drinking water extraction, often by long tunnels. Locally, in the area of the Idstein depression, young extensional faults also formed a high permeable fractured aquifer. Especially at the edges of the graben, it has higher

permeabilities and storage volumes compared to adjacent areas, although the fractured aquifer consist of the same schists and quartzites. Highly mineralized deep waters rise by $CO_2$-gaslift and form natural mineral springs in fault zones surrounding Oberselters and Niederselters (Carlé, 1975). While Na-Cl dominated mineral waters rise along the southern Taunus Border Fault, the mineral waters within the Rhenish Massif contain higher proportions of Ca, Mg and $HCO_3$ and are classified as local formations (Carlé, 1958, Thews, 1972, Kirnbauer et al., 2012). The acidulous waters of Selters in the Taunus or the rather low

mineralized healing waters of Bad Schwalbach or Schlangenbad are among these (Carlé, 1975, Käß and Käß, 2008). Further north, e.g., in Leun-Biskirchen, waters occur that contain a lot of Ca and Mg. In nearby Löhnberg-Selters, however, Na-Ca-Cl water is extracted.

Lower Carboniferous schists form low-permeability fractured aquifers in the region of the rivers Lahn and Dill in the central Hessian part of the Rhenish Massif. A special characteristic is the use of groundwater collected from old drainage tunnels of





abandoned haematite ore mines. They tap large-scale catchment areas where diabases (meta-volcanites), greenstones, greywackes and slates are predominant. Of greater hydrogeological and water management importance are the Middle Devonian massive reef limestones, which represent highly permeable fractured and karst aquifers with high storage volumes, e.g., in the Limburg area.

The relatively well permeable Tertiary volcanic (basaltic) rocks of the Westerwald lie in the west of the Lahn and Dill area on top of the Palaeozoic or its Tertiary clayey weathering surface.

### 1.2.5 Crystalline Odenwald (HYR-10)

The crystalline Odenwald in southern Hesse consists of plutonic igneous and metamorphic rocks – namely granites, diorites,
amphibolites, gneisses – which are referred to as fracture aquifers. Significant permeabilities generally only occur in near-surface weathering zones up to a depth of a few tens of metres. Otherwise, the basement is very low to extremely low permeable (Becht et al., 2017). This also applies to the narrow Hessian strip of the crystalline Spessart. In some areas, a weathering layer of rubble is deposited over the bedrock, in which a pore aquifer of medium to moderate permeability is developed. In the north, Permian Rotliegend of the Sprendlinger Horst covers the crystalline Odenwald. It contains conglomerates, arkoses, sandstones,
mudstones and subordinated limestones and embedded volcanites. Small-scale isolated fracture aquifers, developed mainly in the sandstones and volcanites, are low to very low permeable, only.

## 2 Methods

### 2.1 Criteria for input data

The HLNUG runs a public data base on groundwater and drinking water protection called 'GruSchu' (GruSchu 2020). 'GruSchu' includes data of groundwater, raw water, drinking water and curative water provided by waterworks as well as the public measuring station network. Therefore, 'GruSchu' contains mainly information on shallow aquifers. For deeper aquifers it is thus necessary to compile a complementary data base on natural mineral, thermal and deep groundwaters. To be included, datasets had to comprise the concentration of major elements ($Na^+$, $Ca^{2+}$, $Mg^{2+}$, $K^+$, $Cl^-$, $HCO_3^-$ and $SO_4^{2-}$) and meet at least
one of the following criteria:

- water temperature of more than 20 °C (German definition of thermal water, Himstedt et al., 1907, Michel, 1997)
- concentration of total dissolved solids or free carbon dioxide of at least 1 g $l^{-1}$ (traditional German definition of mineral water, Himstedt et al., 1907, Michel, 1997)
- groundwater origin at least 100 m (definition of the formation water database of the BGR)

Gas composition and isotope data, mainly $\Delta^{13}C$, $^{14}C$ and $\Delta^{34}S$, were included even without concentrations of major elements. Nine analyses (#439, 440, 463, 464, 479, 480, 781, 928, 975) located few kilometers beyond the federal border of Hesse were included to improve the spatial data distribution. Additionally, three data sets are added although they do not meet the criteria mentioned above. The Sossenheimer Sprudel (#544) is included due to its location within Frankfurt and its historical



significance. Wells Groß-Umstadt V (#615) and Mörlenbach Weihrich IV (#772) are nearly 100 m deep and represent the crystalline basement of the Odenwald, where data is generally sparse.

The data sets mentioned above are further supplemented by 18 data sets with mean values for rock units occurring in Hesse from Ludwig (2013). Furthermore, 37 datasets from the Baden-Württemberg, Rhineland-Palatinate or French part of the Upper
Rhine Graben (URG) from Stober & Jodocy (2011), Stober & Bucher (2015) and Sanjuan et al. (2016) were included since only sparse data from literature are available in Hessen for the hydrogeological units of the graben fill of the URG and the Mesozoic to Paleozoic units below.

## 2.2 Literature survey

The database is based almost exclusively on published data archived in the libraries of the TU Darmstadt as well as the BGR. Major sources were the official explanations of geological maps 1 : 25 000 and periodicals of the HLNUG and its precursor organizations ('Geologisches Jahrbuch Hessen', 'Geologische Abhandlungen Hessen', 'Abhandlungen des hessischen Landesamtes für Bodenforschung', and 'Notizblatt des hessischen Landesamtes für Bodenforschung'). Of those, the official hydrogeological map series of Hesse 1 : 300 000 by Diederich et al. (1991) is by far the one which provided most data. Another
main source were three standard works on mineral and thermal waters of Germany and central Europa (Himstedt et al., 1907, Carlé, 1975, Käß and Käß, 2008). Other books or journal articles, published technical reports, theses, own works as well as information on municipal web-pages served as secondary data source. Some data were taken from 'GruSchu', especially to complement hydrochemical analyses to isotope data.

## 2.3 Old analyses

Analyses from the 19th century and the beginning of the 20th century are integrated, if the analysis is of interest for longer time series, no younger analytical data is available or they appear trustworthy due to the experience or reputation of a laboratory or an author, e.g., Justus von Liebig (Liebig, 1839a, 1839c) or Remigius Fresenius (Fresenius, 1886, 1887). Unfortunately, some old analyses had to be excluded, because a conversion into contemporary units or species used today (Table 1, cf. Käß and Käß, 2008) could not be carried out correctly or the datasets were too incomplete (e.g. Liebig, 1839b, Jochheim, 1858, or
Chelius, 1905). Concentrations in old analyses were often given as salts or acids and not as ions or elements like today. Some authors presented both (e.g. Himstedt et al., 1907), but normally salt or acid concentrations had to be converted using molar masses. Since that time, approved molar masses shifted slightly causing a mistake in our recalculation which might be in the range or below the measurement uncertainty in former times and thus pose no problem here.


## 2.4 Explanation of database structure

The database is structured in eleven domains: *metadata*, *references*, *physical-chemical (sum-)parameters*, *chemical parameters*, divided in *major elements*, *minor elements*, and *trace elements*, *dissolved gases*, *free gases*, *isotopes*, as well as *evaluation* and *electrical balance*, available in the excel-version, only (Schäffer et al., 2020).



*'Metadata'* contains information on the location, name, position, altitude, impoundment, geology and water type. Light grey shaded fields in the columns referring to coordinates, altitude, final depth and geology mark own additions of the data set which were not given in the original reference. Coordinates are given in both traditional German Gauß-Krüger coordinate system (transverse Mercator projection, DHDN3) and the modern Universal Transverse Mercator (UTM, WGS84) coordinate

system, respectively. Table 2 summarizes generalized rock types of the aquifer units, which were chosen in accordance to the PetroPhysical Property Database P³ (Bär et al., 2020). Local stratigraphic units and their geological period are chosen in accordance to the stratigraphic table of Germany (DSK, 2016) and the geological overview map of Hesse 1 : 300 000 (HLUG, 2007). Entries in the field of water type (own information) provide a quick overview of the groundwater chemistry. T, M, B and S stands for thermal water, mineral water, brine and acidulous water, classified in accordance to the definition mentioned

in Section 2.1. Brines are defined here using the common balneological definition in Germany of at least 14 g l$^{-1}$ dissolved sodium and chloride (Adam et al., 2000). Cations and anions are listed with descending ratios, if they their equivalent concentration reaches at least 20 % (Michel, 1997).

The domain *'references'* contains information on the date of analysis, the laboratory or person who executed the analysis and a citation. If a data set was found several times in literature, often with complementary information, multiple citations are

given sorted from old to young. The number of analyses for each spring or well is also given sorted from old to young. Dark grey shaded fields indicate comments or explanations available within the observation column, for example, additional information, contradicting values in different literature sources or missing data.

The domains *'physical-chemical (sum-)parameters'*, *'chemical parameters'* contain the most important parameters to classify groundwaters. If authors stated that certain ions or elements were measured, but found to be below the detection limit they

specified, this information is included in the database. Light grey shaded fields in columns concerning the electrical balance and major elements mark own calculations, e.g., bicarbonate from the carbonate hardness or single ions from the electrical balance itself. Total dissolved solids (TDS) are calculated from the sum of all stated concentrations for major, minor and trace elements. The electrical balance considers the concentrations of Na$^+$, K$^+$, Mg$^{2+}$, Ca$^{2+}$, Mn, Fe, HCO$_3^-$, NO$_3^-$, SO$_4^{2-}$ as well as Cl$^-$, and is calculated in accordance to the German standard DIN 38402-62 (2014), given in Eq. (1). This includes a factor of

0.5 in the denominator resulting in twice as high deviation of the electrical balance in comparison to the international common formula (e.g. Appelo and Postma, 2007). The electrical balance of 24 analyses deviate more than ±5 % and is marked with a red filling.

$$\Delta IB \;=\; \frac{\sum c_{eq}(\text{Kat}) - \sum c_{eq}(\text{An})}{(\sum c_{eq}(\text{Kat}) + \sum c_{eq}(\text{An})) \cdot 0.5} \;\cdot\; 100 \quad (\%) \qquad\qquad \text{Eq. (1)}$$




## 3 Results

The database includes 1035 datasets from 560 different measurement points in the entire Hessian territory and adjacent areas (Figure 1). 890 datasets include concentrations of major elements and therefore TDS and electrical balance. The database covers a period of more than 200 years. An analysis of the Schwefelquelle ('Sulfur spring', #520) in Flörsheim from 1810 is the oldest dataset (Thews, 1969). Figure 2 shows the data distribution with respect to depth, temperature, TDS and carbon dioxide content. The depth is classified in four groups (Figure 2a): 24.6 % of the datasets have a depth of 0 to 10 m and are springs, shafts or very shallow boreholes tapping the uppermost aquifer. They are in many cases related to fault zones and therefore releasing water of a deeper origin. Another 30.6 % of the datasets belong to the second group (depth >10 to 100 m), most of them being wells used for drinking water supply. A limit of 100 m is defined here is in accordance to the formation water database of the BGR. The third group (depth >100 to 400 m) is with 37.3 % the largest one. These are mainly wells to supply drinking water, thermal baths or spas, where the limit of 400 m follows the definition of shallow geothermal systems (e.g. PK Tiefe Geothermie, 2007; Sass et al., 2016). The fourth group (deeper than 400 m) consist of 6.9 % of the datasets and are mostly deep spa wells, research drillings or boreholes from hydrocarbon exploration. Borehole Pfungstadt I (#853), southwest of Darmstadt, is the well with the deepest hydrochemical analysis (2,291 m). In 0.6 % of the datasets no information on the depth is available.

In total, 12.7 % of the datasets are thermal waters, 70.6 % are below 20 °C (Figure 2b). The remaining 16.7 % do not have an entry for temperature. A dataset from Kochbrunnen ('Cooking well') in Wiesbaden has a temperature of 67.3 °C and represents the hottest natural thermal water of Hesse. It has 12 entries from 1849 to 2011 (#997-1008) and therefore has the largest number of analyses for one sampling location within the database.

The majority of the datasets, 53.8 %, fulfill the definition of mineral waters due to their TDS, another 11.3 % has sufficient sodium and chloride concentration to be classified as brine in a balneological sense (Figure 2c). Solebohrung III ('Brine wellbore') in Bad Karlshafen (#59), in the northernmost part of Hesse north of Kassel, shows by far the highest TDS with 301 g l$^{-1}$. This extreme high concentration is caused by its origin where the brine leaches halite rocks of the Zechstein. Note that 14.0 % cannot be defined, due to missing hydrochemical data.

In total, 36.5 % of the datasets fulfill the definition of mineral waters due to their free carbon dioxide content, also called acidulous waters (Figure 2d). Georg-Viktor-Quelle B (#345) in Bad Wildungen, located between Marburg and Kassel, shows the highest content of free carbon dioxide (< 21.5 mg l$^{-1}$). 18.8 % of the datasets have no entries on the carbon dioxide content.

The first step of the statistical evaluation was to match datasets with the individual hydrogeological units of Hesse (see Section 1.2) based on their location (i.e. coordinates). The second step was to prepare and condition the bulk database itself. The conditioning included the following:

1. Delete datasets with incomplete entries (e.g. missing hydrochemical data);
2. Reduce multiple datasets to only one measurement per location (no time series considered);
3. Delete measurements with electrical balance exceeding an error greater ± 5%.





After the conditioning, the database comprised 545 individual datasets. For the statistical evaluation the parameters *temperature*, *pH* and *TDS* as well as major ions ($Na^+$, $K^+$, $Ca^{2+}$, $Mg^{2+}$, $Cl^-$, $HCO_3^-$ and $SO_4^{2-}$) were considered.

In total, 215 datasets describe the Upper Rhine Graben and Mainz Basin (HYR-03) including Vogelsberg volcanites (HYR-03_VV; n=7). The pH values of water samples from this group range from 5.7 to 8.2. The average is 6.9 with a standard deviation of 0.6. At 18 locations, the groundwater pH is below 6. These originate from Silurian gneisses and phyllites, Devonian limestones and quartzites as well as unconsolidated rocks of the Quaternary. Temperatures within HYR-03 are between 6.0 and 66.3 °C (average=16.2; standard deviation=12.2). Note that as many as 50 datasets do not have an entry for temperature. Twenty-one datasets have groundwater temperatures above 20.0 °C. All but one of these groundwaters show $Na^+$ as the dominating cation, most often in combination with $Cl^-$ (pure Na-Cl water type at 14 locations, one location has a Ca-Na-SO$_4$ water type). Within HYR-03, the highest groundwater temperatures above 40 °C mainly occur in Silurian gneisses and phyllites of Wiesbaden. By plotting groundwater temperature versus depth (Figure 3), several distinct subgroups can be differentiated. The largest subgroup is characterized by mixed groundwater due to convection. Convection occurs when hydraulically permeable subsurface pathways, e.g., faults or fractures, allow deeper, connate groundwater to migrate upwards. In the shallower subsurface (up to approximately 250 m bsl), this fossil, typically warm or hot and mineralized groundwater mixes with dilute and colder groundwater of meteoric origin (e.g. Schäffer, 2018). The subgroup "normal" represents groundwater with temperatures roughly following the typical Hessian geothermal gradient (green). One subgroup shows increased groundwater temperatures at or close to the surface (hot, yellow) and the subgroup "very hot" shows groundwater temperatures ≥ 60 °C. One groundwater is characterized by cold temperatures well below the geothermal gradient (#673, Solebohrung I in Kassel-Bad-Wilhelmshöhe). Here, the groundwater temperature is 14.9 °C while the borehole depth is 1,316 m bsl.

Groundwater mineralizations (i.e. TDS) of HYR-03 range from 72 to as much as 122,661 mg l$^{-1}$ (Table 3). The groundwater from location #673 (see above) has the maximum TDS with a pronounced Na-Cl water type. This is noteworthy as from this site no halite deposits are known which may explain the high TDS and Na-Cl domination. Because this value is approximately twice as high as the second highest (64,396 mg l$^{-1}$) it is considered as an outlier and hence excluded from further statistics. Groundwater with TDS concentrations above 10,000 mg l$^{-1}$ (n=29) show a pronounced Na-Cl dominance. These high mineralization mainly occur in unconsolidated rocks of the Neogene (n=10) as well as in Permian Zechstein (n=5) and Devonian limestones (n=4). TDS concentrations below 1,000 mg l$^{-1}$ are dominated by $Ca^{2+}$ and $HCO_3^-$. These mainly occur in groundwater from Triassic Buntsandstein, Quaternary and Tertiary aquifers. No depth-TDS correlation is apparent.

Groundwater from the Vogelsberg volcanites (HYR-03_VV; n=7) are characterized by increased pH values between 8.0 and 9.9, low mineralizations between 151 and 256 mg l$^{-1}$ (Table 3). Chemically, $Na^+$ and $HCO_3^-$ are the major dissolved ions, and the groundwater of HYR-03_VV represents Na-HCO$_3$-Cl water types (one Ca-Mg-HCO$_3$).



The Central German Basin Range (HYR-05) comprises 195 datasets. Groundwater temperatures are between 7.0 and 50.0 °C. Note that 22 datasets have no temperature entries. At seven locations groundwater temperatures are above 20 °C. These originate from depths greater 100 m bsl and mainly occur in Permian formations. Only the location with the highest temperature (50.0 °C) originates from the Silurian Lorsbach-Formation (gneiss, metarhyolite). In Figure 4 depth-temperature dependence for the groundwater of HYR-05 are displayed. While a large subgroup of groundwater is characterized by convection (purple) and another subgroup roughly follows the typical Hessian geothermal gradient (normal, green) similar to groundwater of HYR-03, two new features are apparent. There is one outlier with 50.0 °C at the surface, and one subgroup with groundwater temperatures below the geothermal gradient (cold, blue).

The pH values of samples from HYR-05 are between 5.2 and 9.3 (average = 6.8; standard deviation = 0.7). Increased pH values above eight (n=4) are found in groundwater from Permian Rotliegend and Rhön volcanite. These datasets are dominated by $Na^+$ and $HCO_3^-$. Lower pH values below six typically occur in Triassic sandstones and Permian rock series. Note that Schlossquelle in Bad Wildungen (#361) has a pH of 6.0 and a $Mg$-$Ca$-$Na$-$HCO_3$ water type reflecting a $Mg^{2+}$ domination. A total number of 70 datasets do not have an entry for pH. Mineralization ranges from 65.7 to 300,983 mg l$^{-1}$ (Table 3). The average is 10,723 mg l$^{-1}$ (note the median of 1,613 here) with a standard deviation of 28,629 mg l$^{-1}$. Hence, HYR-05 has the highest TDS average of the five major hydrogeological regions of Hesse. Groundwater with TDS contents above 10,000 mg l$^{-1}$ (n=40) are all characterized by Na-Cl water types. At TDS values above 50,000 mg l$^{-1}$ (n=9) groundwater mainly occur in Zechstein formations. Lower mineralization below 1,000 mg l$^{-1}$ (n=79) relate to groundwater from sandstone of the Buntsandstein. These groundwaters are characterized by $Ca$-$Mg$-$HCO_3$ water types. A correlation of TDS with depth is not evident for HYR-05, whereas in northern and eastern Hesse deep waters from the Buntsandstein are usually mixed by ascending leachate waters of the Zechstein, especially next to the salt slope.

The Southern German Cuesta Landscape in south-eastern Hesse (HYR-06) comprises 15 groundwater samples mainly from Buntsandstein and Zechstein formations. Temperatures are between 10.0 and 20.5 °C (four datasets do not have temperature values). Only one groundwater has a temperature above 20.0 °C, the remainder is between 10.0 and 14.0 °C. pH values range from 6.0 to 7.3, with seven samples having no pH entry. Mineralization ranges from 34.8 to 27,176.0 mg l$^{-1}$ (Table 3). The average TDS is 5,524 mg l$^{-1}$ and the standard deviation is 8,621 mg l$^{-1}$. Lowest mineralization occurs in groundwater from formations of the Middle Buntsandstein. These typically have a $Ca$-$HCO_3$(-Cl) water type. TDS values above 10,000 mg l$^{-1}$ are Na-Cl dominated; the Martinusquelle in Bad Orb additionally shows a significant $SO_4^{2-}$ influence of 2,100 mg l$^{-1}$. Here it is noteworthy that in Bad König three samples show a $Ca$-$K$-$HCO_3$-Cl water type. These are the only datasets of the entire database with $K^+$ as the co-dominating cation. Among the thermal waters of Hesse, Bad König is the only location where thermal water from granites is extracted. These granites - presumably mica (i.e. Muscovite) - may explain the high content of dissolved $K^+$.



The Rhenish Massif in western and north-western Hesse (HYR-08) comprises 115 datasets in total. The depth interval is between 0 and 1,000 m bsl. Groundwater temperatures within HYR-08 range from 6.2 to 66.2 °C; average=15.1 °C with a standard deviation of 10.0 (eight times n.d.). By far the highest temperatures ≥49 °C are from groundwater in Wiesbaden (Figure 5, subgroup "very hot"). Increased temperatures between 20.0 and 34.3 °C are mainly found both in Silurian-Ordovician phyllites and gneisses (n=9) as well as in Devonian rocks (n=4). Temperatures below 10.0 °C are typical for Devonian schists (e.g., Hunsrück and Wissenbach schist). While the majority of groundwater belong to the subgroup "convection", one subgroup shows increased groundwater temperatures at or close to the surface ("hot"). Note the subgroup "very hot" with groundwater temperatures ≥50 °C. Analogous to HYR-05, a subgroup with groundwater temperatures below the geothermal gradient ("cold") is apparent in Figure 5.

The pH values of HYR-08 are between 5.4 and 8.0; the average is 6.2. Note that 53 datasets do not have a pH entry. As much as 35 datasets show a pH value below six. These mainly belong to Silurian-Ordivician gneisses and phyllites (n=18) and Devonian schists (n=12), respectively.

Groundwater mineralization within HYR-08 range from 53.0 to 26,471.9 mg l$^{-1}$. The average TDS is 4,601.6 mg l$^{-1}$ with a standard deviation of 5,124.3 mg l$^{-1}$ (Table 3). It is noteworthy here that location #509 has the lowest mineralization of 53.0 mg l$^{-1}$ of which 16.8 mg l$^{-1}$ are $H_2SiO_3$. The highest mineralization above 10,000 mg l$^{-1}$ mainly occurs in groundwater from Silurian metavolcanites (n=16) as well as Silurian-Ordovician phyllites (n=7). These groundwaters show a pronounced Na-Cl water type. The lowest mineralisation below 1,000 mg l$^{-1}$ are Ca-Mg-HCO$_3$ dominated. These low TDS are typical for groundwater of Devonian schists and Triassic-Devonian sandstones. Three datasets show groundwater with $SO_4^{2-}$ as the main anion. These are from the Permian (Zechstein and Rotliegend). No correlation with depth is apparent for TDS.

The crystalline Odenwald in south-eastern Hesse as part of the south-western German basement (HYR-10) comprises five datasets; one from Rotliegend (arkose), Zechstein (?) and Paleogene (marl), as well as two from sandstones of the Buntsandstein (S3 and S3-S6). Groundwater temperatures range from 10.9 to 18.0 °C (T$_{max}$ from Rotliegend). pH values are between 6.7 and 7.3 (two datasets without entry). Compared with the other major hydrogeological regions of Hesse, mineralization of HYR-10 are rather low ranging from 277 to 2,970 mg l$^{-1}$ (Table 3). It is noteworthy that all the five groundwaters have different water types.

**4 Data availability**

The database of mineral, thermal and deep groundwaters of Hesse has been made available by Schäffer at al. (2020) as xlsx and csv-file at TUdatalib, https://doi.org/10.25534/tudatalib-340.



## 5 Conclusion and Outlook

The hydrochemical composition of groundwaters and related fluid properties are of great importance for the planning and trouble-free operation of geothermal power plants or direct heat utilisation as well as to understand hydrochemical processes in aquifers. Here, we describe the first publicly available data compilation of thermal, mineral and deep groundwater of Hesse. We compiled and evaluated hydrochemical data in a database for all geological units that are assessed as suitable for deep geothermal use. Through extensive literature research, it was possible to compile a data set that thoroughly covers the Hessian state territory, both spatially and temporally. In the near future, a major improvement of data accessibility is expected after implementation of the 'Geologiedatengesetz' (Geology Data Act) passed in 2020. It transforms private into public data and allows state authorities to publish data which has been rated as classified so far. The HLNUG has thousands of documents, reports, statements, etc., which could not be evaluated and used up to now due to confidentiality and property rights. Thus, an update and extension of the database is planned, and contributions of other people are welcome.

The database can serve as a basis or tool for other projects or scientific questions. It can be used to estimate reservoir temperatures with solution geothermometers. In the best case, geothermometers can be used to determine reservoir temperatures for individual aquifers of the entire Hessian territory. The database can be also used to re-evaluate the genesis and circulation of deep groundwaters. In addition, the isotope data can be used for calculating groundwater ages or evaluating residence times, which in turn are a criterion for the search of a repository for radioactive waste (StandAG, 2017). Furthermore, the hydrochemical database will be integrated by the BGR into the formation water database covering the entire German territory.

## 6 Author Contributions

IS was involved in the conceptualization and funding acquisition. JGF organized data from HLNUG and wrote chapter 2. KB was responsible for conceptualization, funding acquisition and project administration and wrote section 1.1. RS collected data from the library of the TU Darmstadt, compiled the data and developed the database, wrote chapters 2, 4, 5 and contributed to chapter 3. SF collected data from the library of the BGR, did the formal analysis, wrote chapter 3 as well as contributed to section 1.1. All authors conducted an internal review of the manuscript.

## 7 Conflict of interest

The authors declare that they have no conflict of interest.



# 8 Acknowledgements

This publication is part of the research project '3D modelling of the petrothermal and medium deep geothermal resources for power production, direct heat utilization and storage of the federal state of Hesse' funded by the Federal Ministry for Economic Affairs and Energy (BMWi), grant no. 0325944.

Lena Muhl, Sofia Nalbadi and Petra Kraft helped to collect and review the literature, transferred the data or own sources into the database and suggested restructures of the database where needed. Renate Senner (HLNUG) gave useful hints on the groundwater data base of Hesse (GruSchu). Christoph Kludt (HLNUG) provided a data collection of selected mineral and medicinal waters of Hesse. We also appreciate help and support of Marion Homann and the library team at GeoZentrum Hannover (i.e. BGR) during the data search. We thank all contributors and also our external cooperation partners for their

support and work to fill the database with valuable data or by providing valuable reports or publications to be included in the compilation.

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



**Figure 1: Hydrochemical sampling points and hydrochemical regions in the federal state of Hesse. Abbreviation of cities: AB – Aschaffenburg, DA – Darmstadt, F – Frankfurt, FD – Fulda, GI – Gießen, GÖ – Göttingen, HD – Heidelberg, HU – Hanau, KL – Kaiserslautern, KO – Koblenz, KS – Kassel, MA – Mannheim, MR – Marburg, MZ – Mainz, SI – Siegen, WI – Wiesbaden. Generalized geological map after Schäffer et al. (2018).**




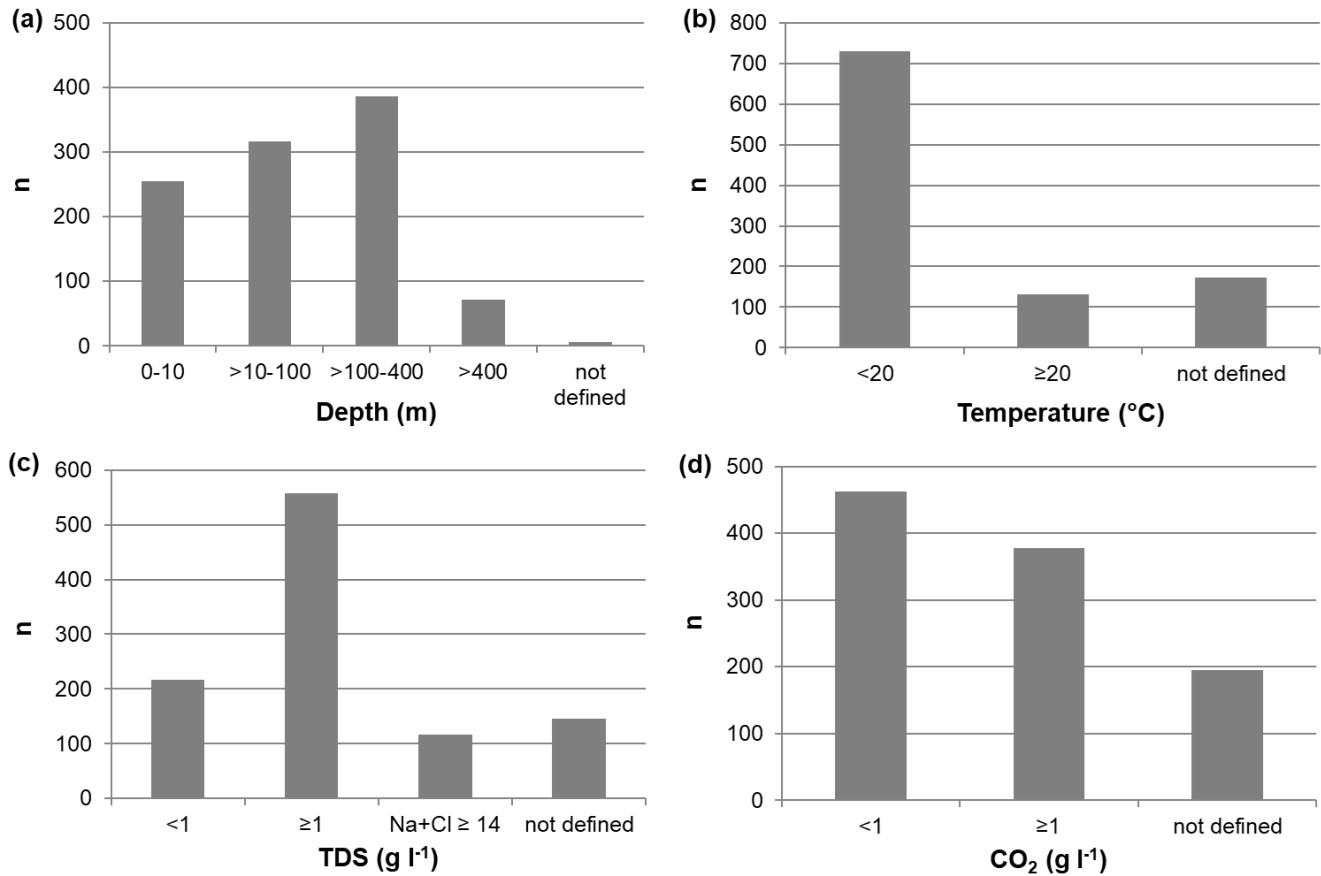

**Figure 2: Data distribution with respect to (a) depth of the screened section of the water well, (b) water temperature, (c) concentration of total dissolved solids (TDS) or sodium plus chloride, and (d) the content of free carbon dioxide.**

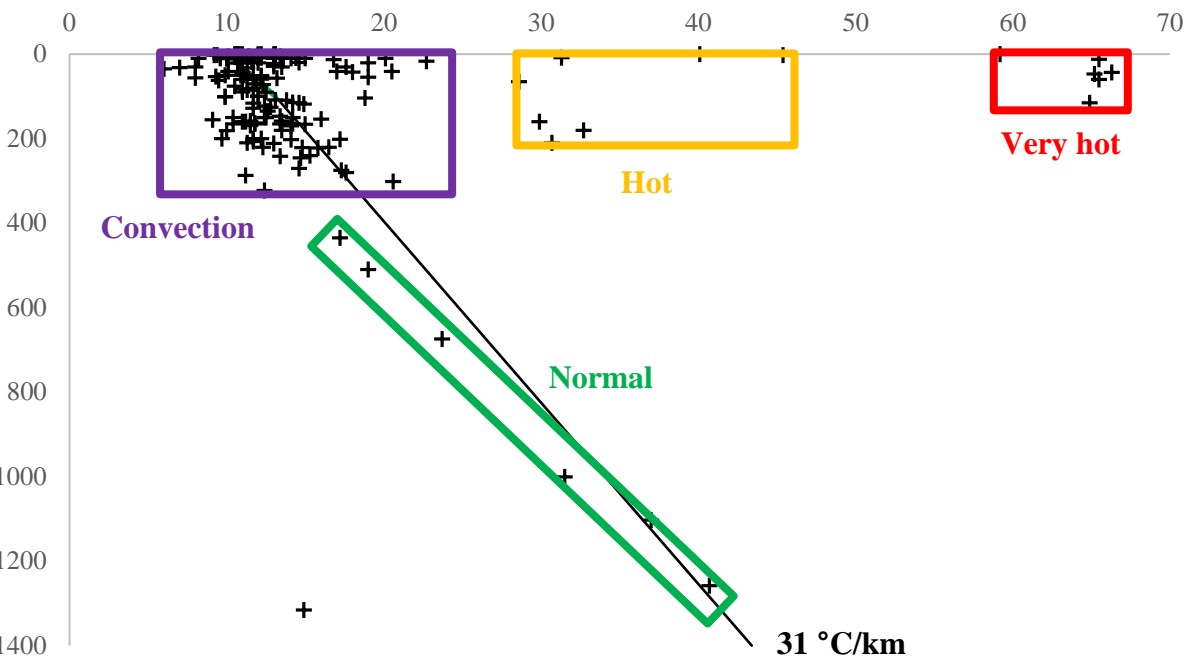

**Figure 3: Temperature data of HYR-03 plotted against depth. While the majority of data are characterized by "convection" and/or follow the typical Hessian geothermal gradient ("normal"), also two data clusters of "hot" and "very hot" groundwaters are apparent. Note that the field "convection" has been constrained rather arbitrary and not based on firm scientific evidence.**



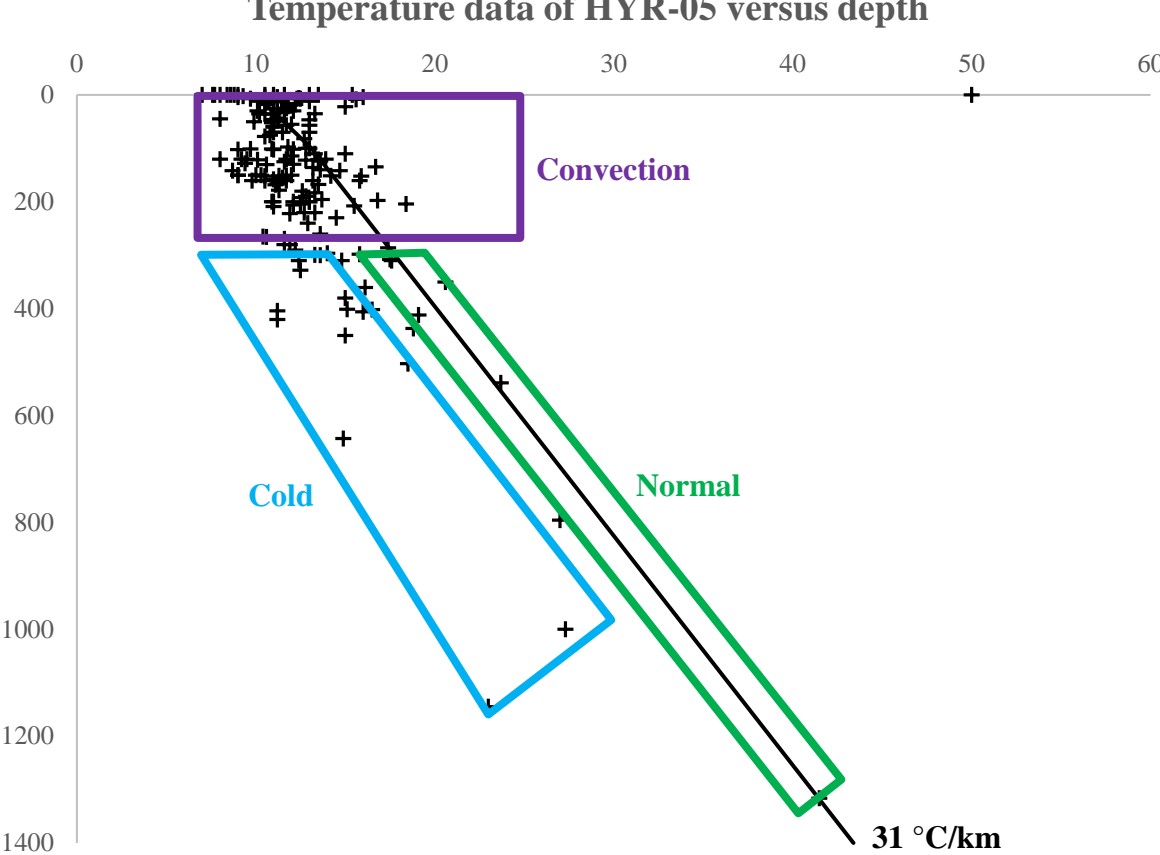

**Figure 4: Temperature data of HYR-05 plotted against depth. Similar to HYR-03 (Figure 3) the majority of data from HYR-05 are also characterized by "convection" and/or follow the typical Hessian geothermal gradient. However, as opposed to HYR-03, neither "hot", nor "very hot" groundwater clusters are apparent. Instead, HYR-05 shows a data cluster of "cold" groundwaters with temperatures below the typical Hessian geothermal gradient. Note that the field "convection" has been constrained rather arbitrary and not based on firm scientific evidence.**



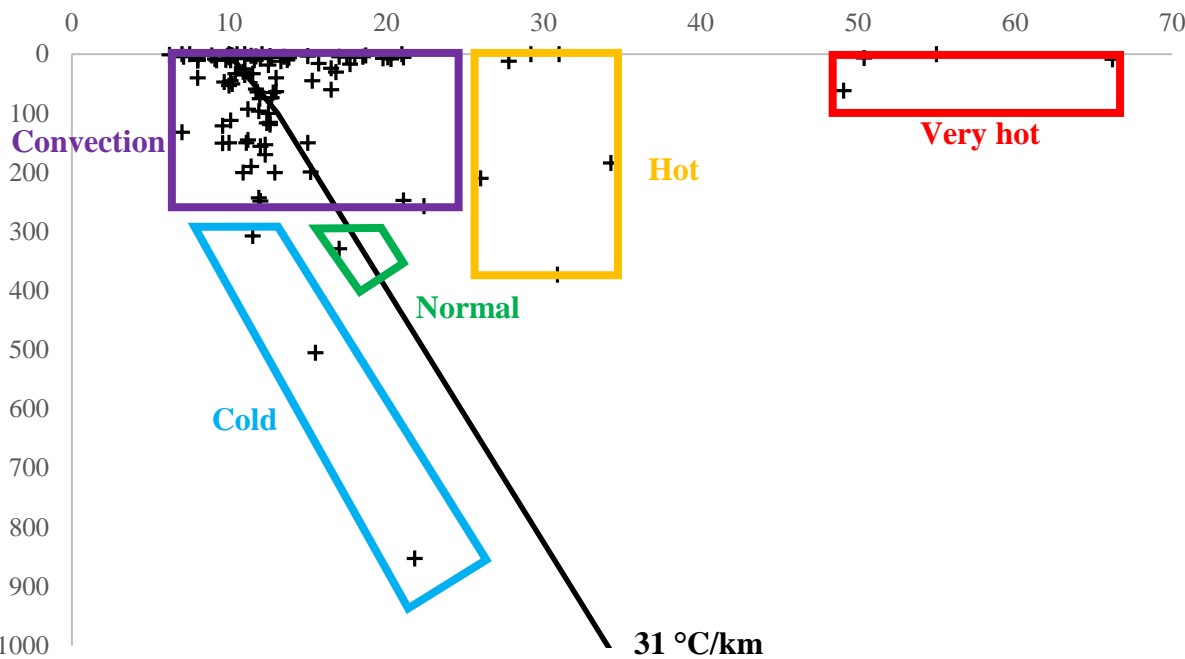

**Figure 5: Temperature data of HYR-08 plotted against depth. Next to the clusters "hot" and "very hot" HYR-08 also shows the cluster "cold". Only one groundwater (below 250 m bsl) shows a "normal" temperature in accordance with the typical Hessian geothermal gradient. Note that the field "convection" has been constrained rather arbitrary and not based on firm scientific evidence.**





**Table 1: Conversion of obsolete units into contemporary units, or species preferentially used today in water analysis (cf. Käß and Käß, 2008).**

| From | To | Factor |
|---|---|---|
| Réaumur-Temperatur (°R, °Re or °Ré) | Celsius-Temperature (°C) | 1.25 |
| Electric conductivity (mS/m) | Electric conductivity (µS/cm) | 10 |
| Electric conductivity at 20 °C (µS/cm) | Electric conductivity at 25 °C (µS/cm) | 1.116 |
| Carbonate hardness (°dH) | $HCO_3^-$ (mg l$^{-1}$) | 21.76 |
| $SiO_2$ (mg l$^{-1}$) | $H_2SiO_3$ (mg l$^{-1}$) | 1.3 |
| Si (mg l$^{-1}$) | $H_2SiO_3$ (mg l$^{-1}$) | 2.78 |
| B (mg l$^{-1}$) | $HBO_2$ (mg l$^{-1}$) | 4.052 |
| $HAsO_4^{2-}$ (µg l$^{-1}$) | As (µg l$^{-1}$) | 0.536 |
| $H_2TiO_3$ (µg l$^{-1}$) | Ti (µg l$^{-1}$) | 0.489 |
| $HS^-$ (mg l$^{-1}$) | $H_2S$ (mg l$^{-1}$) | 1.030 |
| Radioactivity (Ci) | Radioactivity (Bq l$^{-1}$) | $3.67 \cdot 10^{10}$ |
| Radioactivity (ME) | Radioactivity (Bq l$^{-1}$) | 13.47 |



**Table 2: Rock types, local stratigraphic units and geological periods after HLUG (2007) and DSK (2016) used for classification in the hydrochemical database.**

| Rock type | Local stratigraphic unit | | Geological period |
|---|---|---|---|
| Unconsolidated rock, carbonate bearing silt | Pleistocene | | Quaternary |
| | Pliocene | | |
| Basalt, tuff, hawaiite | Westerwald volcanite | Miocene | Neogene |
| | Rhön volcanite | | |
| | Vogelsberg volcanite | | |
| Unconsolidated rock, marl | Frankfurt-Fm., Wiesbaden-Fm. | | |
| Unconsolidated rock, marl, limestone | Rockenberg-Formation | Oligocene/ Miocene | Neogene/Paleogene |
| | Cerithien-Layers | | |
| Unconsolidated rock, marl | Cyrenenmergel-Group | Oligocene | Paleogene |
| | Rupelton-Fm., Bodenheim-Fm. | | |
| | Pechelbronn-Formation | | |
| Unconsolidated rock | Priabonian | Eocene | |
| Dolomite | Keuper | | Triassic |
| Limestone | Muschelkalk | | |
| Sandstone, gypsum | Röt-Formation | Upper Buntsandstein | |
| Sandstone | Solling-Formation | Middle Buntsandstein | |
| | Hardegsen-Formation | | |
| | Detfurth-Formation | | |
| | Volpriehausen-Formation | | |
| Sandstone | Bernburg-Formation | Lower Buntsandstein | |
| | Calvörde-Formation | | |
| Pelite, sandstone | Ohre-Fm., Friesland-Fm., Fulda-Fm. („Bröckelschiefer-Formation') | Zechstein | Permian |
| Dolomite, limestone, marl, pelite, halite rock, conglomerate, anhydrite | Aller-Formation | | |
| | Leine-Formation | | |
| | Staßfurt-Formation | | |
| | Werra-Formation | | |
| Pelite, sandstone, conglomerate, arkose, | Rotliegend | | |
| Graywacke, limestone, conglomerate, dolerite | Dainrode-Formation | | Carboniferous |
| Slate | Lelbach-Formation | | |
| Chert | Bromberg-Formation, Laisa-Formation, Hardt-Fm. | | |
| Slate, graywacke, dolerite | Metavolcanite | | |
| Granite, granodiorite | Odenwald plutonic rocks | | |
| Limestone, slate, sandstone | Adorf-Layers | | Devonian |
| Limestone, dolomite | Massenkalk ('massive limestone') | | |
| Dolerite, tuff | Schalstein ('peel stone') | | |
| Slate, quartzite | Wissenbach-Schist | | |
| Quartzite, slate, sandstone | Ems-Quartzite | | |
| Slate | Hunsrück-Schist | | |
| Quartzite, slate | Taunus-Quartzite | | |
| Quartzite, sandstone | Hermeskeil-Layers | | |
| Sandstone, slate | Bunte Schiefer ('colourful slate') | | |
| Muscovite-Biotite-Gneiss | Böllstein-Gneiss | | Silurian |
| Gneiss, greenschist | Metarhyolite | | |
| Phyllite | Lorsbach-Formation | | Silurian/Ordovician |



**Table 3: Specific statistical values of major ions and TDS separately listed for the individual hydrogeological regions of Hesse.**

| | | Major hydrogeological region of Hesse [n=545] | | | | | |
|---|---|---|---|---|---|---|---|
| | | HYR-03 [n=208] | HYR-03_VV [n=7] | HYR-05 [n=195] | HYR-06 [n=15] | HYR-08 [n=115] | HYR-10 [n=5] |
| $Na^+$ | Min [mg l$^{-1}$] | 3.0 | 6.8 | 0.9 | 1.8 | 1.6 | 5.5 |
| | Max [mg l$^{-1}$] | 43690.0 | 71.1 | 118463.0 | 8894.0 | 7595.0 | 549.0 |
| | Median [mg l$^{-1}$] | 289.0 | 60.2 | 66.0 | 69.8 | 318.0 | 16.2 |
| | Average | 1616.8 | 49.1 | 3368.4 | 1486.8 | 1160.4 | 150.6 |
| | Std dev. | 4191.3 | 23.4 | 10723.4 | 2719.0 | 1662.6 | 233.9 |
| $K^+$ | Min [mg l$^{-1}$] | 0.0 | 0.2 | 0.9 | 1.4 | 0.0 | 2.0 |
| | Max [mg l$^{-1}$] | 4689.0 | 2.0 | 3663.0 | 380.0 | 357.0 | 21.2 |
| | Median [mg l$^{-1}$] | 18.1 | 0.7 | 8.4 | 83.3 | 11.4 | 3.6 |
| | Average | 91.7 | 0.9 | 132.4 | 107.6 | 57.0 | 7.6 |
| | Std dev. | 403.5 | 0.6 | 392.6 | 123.9 | 81.9 | 9.1 |
| $Mg^{2+}$ | Min [mg l$^{-1}$] | 0.0 | 0.5 | 0.0 | 0.8 | 0.8 | 3.7 |
| | Max [mg l$^{-1}$] | 961.6 | 12.1 | 965.8 | 545.0 | 262.0 | 89.2 |
| | Median [mg l$^{-1}$] | 45.0 | 1.5 | 44.1 | 28.9 | 60.0 | 12.1 |
| | Average | 82.3 | 3.1 | 107.3 | 88.7 | 72.1 | 33.4 |
| | Std dev. | 121.5 | 4.1 | 152.1 | 140.6 | 52.5 | 36.3 |
| $Ca^{2+}$ | Min [mg l$^{-1}$] | 4.5 | 4.6 | 0.0 | 5.6 | 5.8 | 4.0 |
| | Max [mg l$^{-1}$] | 2152.0 | 20.7 | 2820.0 | 959.0 | 1377.0 | 262.3 |
| | Median [mg l$^{-1}$] | 193.9 | 5.6 | 168.3 | 114.5 | 209.7 | 64.0 |
| | Average | 287.2 | 7.9 | 369.6 | 289.9 | 262.7 | 101.4 |
| | Std dev. | 328.5 | 5.7 | 488.8 | 334.5 | 224.9 | 98.7 |
| $Cl^-$ | Min [mg l$^{-1}$] | 3.0 | 5.5 | 2.0 | 3.0 | 2.9 | 6.6 |
| | Max [mg l$^{-1}$] | 70970.0 | 31.6 | 174000.0 | 16330.0 | 14020.0 | 1018.0 |
| | Median [mg l$^{-1}$] | 357.2 | 30.3 | 59.9 | 154.4 | 283.7 | 14.5 |
| | Average | 2644.9 | 22.0 | 5357.6 | 2587.9 | 1888.0 | 224.2 |
| | Std dev. | 6964.6 | 11.3 | 16434.6 | 4786.1 | 2832.8 | 444.7 |
| $SO_4^{2-}$ | Min [mg l$^{-1}$] | 0.0 | 0.0 | 0.5 | 0.0 | 0.0 | 9.6 |
| | Max [mg l$^{-1}$] | 5526.0 | 13.5 | 5300.0 | 2100.0 | 2238.0 | 253.8 |
| | Median [mg l$^{-1}$] | 57.2 | 8.7 | 181.0 | 48.7 | 24.0 | 36.7 |
| | Average | 281.7 | 7.8 | 645.0 | 318.9 | 98.9 | 88.0 |
| | Std dev. | 693.1 | 4.6 | 996.6 | 595.9 | 314.3 | 99.4 |
| $HCO_3^-$ | Min [mg l$^{-1}$] | 23.9 | 60.2 | 19.6 | 16.0 | 18.3 | 85.0 |
| | Max [mg l$^{-1}$] | 4000.0 | 138.3 | 4119.7 | 1952.0 | 3070.0 | 712.7 |
| | Median [mg l$^{-1}$] | 546.6 | 132.3 | 341.7 | 461.0 | 860.3 | 393.0 |
| | Average | 750.0 | 112.9 | 641.7 | 622.8 | 967.0 | 390.5 |
| | Std dev. | 688.3 | 34.0 | 713.5 | 607.9 | 664.1 | 222.9 |
| TDS | Min [mg l$^{-1}$] | 71.9 | 150.5 | 65.7 | 34.8 | 53.0 | 277.2 |
| | Max [mg l$^{-1}$] | 122660.7 | 255.5 | 300983.1 | 27176.0 | 26471.9 | 2969.9 |
| | Median [mg l$^{-1}$] | 2071.5 | 240.6 | 1613.1 | 1138.2 | 2580.5 | 598.3 |
| | Average | 5751.7 | 216.8 | 10616.7 | 5524.1 | 4529.0 | 1008.8 |
| | Std dev. | 12148.7 | 45.9 | 28629.8 | 8621.1 | 5144.3 | 1105.2 |