# Peer review of "Mineral, thermal and deep groundwater of Hesse, Germany"

_Earth System Science Data, 2021_

## Author Comment (AC1)

**Reply to comments of Referee #1**

| Page, line | Comment | Reply |
|---|---|---|
| 1, 26/27 | "Almost exclusively from publications" – where is rest of the data from? Consider reformulation. | We added: "with minor supplement by personal communications and own measurements". |
| 1, 28/29 | "reduce uncertainties" – unclear in this context, please reformulate. | We added: "when predicting groundwater properties". |
| 2, 3 | Suggest to rather start like "The physical and chemical properties of groundwater determine its suitability…" | Text modified accordingly |
| 2, 8/9 | suggest to write "… we developed the database … of the federal state of Hesse, Germany." | Text modified accordingly |
| 2, 21 | can you give a reference for the mentioned software? | Done |
| 2, 28 | recommend to generally omit "see" from references to figures, tables etc. | Done |
| 3, 1 | recommend to use the term "Southern German Scarplands" rather than "Cuesta Landscape". It is the more widely used term, at least in my experience. | Cuesta Landscape was replaced by Southern German Scarplands in the entire manuscript. |
| 7, 1 | you talk about 100 m below ground level, I suppose. Please clarify. | Yes, thanks for the suggestions, we added "below ground level". |
| 8, 8 | not sure if "impoundment" is the right word here. May "sealing" or "packing" be more to the point? Please check. | Impoundment is replaced by „water-catchment system". If this term is also regarded improper, we would use „sealing". |
| 9, 5 | Eq. 1: recommend to explain equation variables directly below the equation. | Done. |
| 13, 8 | P13, L8: "temporally" – do you mean in terms of covered stratigraphy? If so, better use that word. | We mean the historic time span of the data starting from 1810, but mostly from the 20$^{th}$ century till this day. We replaced temporally by chronologically. |
| Fig. 1 | I am missing the city of Kaiserslautern in the map… add, or omit from the caption. Also, add a green cross to the legend, or explain it in the caption. | Thanks for the hint, Kaiserslautern is covered by the overview map of Germany and was deleted in the caption. Now, the green crosses are explained in the caption. |
| Fig. 3-5 | Add axis labels! | Done. |
| Fig. 4 | Any explanation for the 50 °C outlier mentioned earlier? | Thanks for this hint. Indeed, this sample was assigned to the wrong hydrogeological region, because it is located very close to the border of HYR-3 and HYR-8, which is defined in this area by the Taunus Border Fault (cf. page 3, lines 20-24). The Taunus Border Fault belongs to a several km wide fault zone. There are numerous tectonic blocks and parallel to subparallel faults. The exact position of single faults and blocks is especially in urban areas unknown. The sample is no longer an outlier after its relocation to HYD-8. Figures 3 and 4 were revised accordingly. |
| Tab. 3 | Please check the used font, it is obviously different from the rest of the article. | Done. |

| Database | In column K, I suggest to concretize "Altitude" – I guess it is the elevation of the drilling/sampling site? | We realized this suggestion in the second release of the database (its column J in the second version). |
|---|---|---|

All technical corrections were implemented.

---

## Author Comment (AC2)

**Reply to comments of Referee #2**

| | | |
|---|---|---|
| 2, 19 | The authors assume that many terms are understood by the readers, but I am not sure this is the case. For example, in Page 2, Line 15 you use the term 'formation water', without explanation. Same is for the term 'mineral water'. I think these terms are not broadly used and their scientific definition may not be clear for many readers. This is the case for many other terms. Please go over the paper and make sure you use the common terms / explain terms which are not trivial. | Definitions were already given in the manuscript, e.g. page 2, lines 20/21 or page 6, line 31 to page 7 line 1. We see no need for further definitions. |
| 3, 10/11 | 'mineralisation increases. What does it mean? increase of EC? TDS? Density? Please clarify. | From our perspective it is clear that TDS and therefore also EC and density increase. To avoid any doubts, we replaced 'mineralisation' by 'TDS' in the first part of the sentence, but maintained 'mineralized waters' in the second part of the sentence. |
| 8, 9 | It is not clear what the 'Light grey' referring to….? | 'Light grey' is referring to certain fields in the database. We believe that this is clear from the context (title of section 2.4 and previous sentences). |

---

## Author Comment (AC3)

* * *
**Reply to comments of Mirko Mälicke (Referee #3)**

| | | |
|---|---|---|
| 2, 20 | Please give the full name of ‚TDS' on the first occurrence. | Done. |
| 2, 21 | As a non-geologist I am not aware of the ‚GeoODin' software and further reference or a brief description would be helpful (maybe a footnote). | We added a reference. |
| 3, 8-18 | This clear description of the geological conditions could further be enhanced by a geological cross-section graph, if available. At least my imagination is not enough to picture this from the text, only. | We agree that one or several geological cross sections would be interesting. At the same time we kindly ask to take into consideration that the effort to produce such graphs is far beyond the scope of this publication. We will take into account this suggestion in future studies. |
| 4, 18-33 | Is this detailed description general knowledge of the area or the result of some kind of geological survey(s), that could be referenced here? | General geological and hydrogeological knowledge as described in the explanations of the respective local geological maps is no cited. However, we added some references to special geological surveys in this section as well as in the first paragraph of the motivation (page 2, lines 6/7), the geology and hydrogeology of Hesse (page 2, lines 26/27) and the explanation of the database structure (page 8, lines 18-20). |
| 7, 3 | For me it is not clear, which ‚nine analyses' are referenced here and where the numbers come from. | We added "in the database" |
| 7, 11 | I think here the German ‚Hessen' is used accidentally, instead of ‚Hesse' | Yes, thanks for the hint, it is corrected. |
| 7, 22 | What exactly is a ‚secondary data source' in this context? Are these sources for auxiliary data/metadata/background information or the data itself? If the latter, I would suggest compiling an overview table of used sources for the appendix. | We mean sources of hydrochemical analyses/data and named them at the beginning of the sentence. We entitled these sources as secondary, because they contribute only with a small share to the database. The main/large sources are described in the sentences before. |
| 8, 8 | Please clarify the difference between ‚location' and ‚position' here. | Clarification inserted in parentheses. |
| 8, 8-10 | If metadata was missing in the original reference, how could this information be added by the authors? Is this compiled from the ‚secondary data sources' (see comment above); if yes I would suggest to clearly make this connection here. | In fact, the completion of missing metadata was very time consuming (sometimes not successful) and required different approaches. Just three examples: 1) Position is given, but altitude is missing: Enter coordinates in digital terrain models and extract the altitude. 2) A location is given (town, valley, hill, pit, street, railway…), but coordinates are not informed. Investigation in old or modern maps and topographic maps to find the exact position or the least receive a good estimation. 3) Specification on geology missing: Investigation in geological maps or geological maps or comparison to neighboring datasets to add information. We believe that a detailed explanation of our approach would be too much. But if you believe this is of interest for the reader than we would add a paragraph here. |

| 8, 15 | Where does the ‚own information' about water type originate from? | We emphazise "own information" here, because there is also a field water type as mentioned by the author. The explanation for the origin of the own classification of the water type is given in the next sentence and a cross reference is given to section 2.1 were the definitions are explained. |
|---|---|---|
| 9, 11 – 10, 4 | The authors present a lot of helpful information about the database here, and how the datasets are distributed. I think it would be helpful to add absolute amounts (of datasets) to the presented percentage numbers. | Done. |
| 9, 24 | From my limited geo-hydrological understanding it is hard for me to imagine a use-case for borehole analysis data with missing depth information. How can this data turn helpful? Also, 0.6%, how many datasets are these? | We added (n = 7). Indeed, missing information on the depth is a major blemish, limiting data evaluation. Nevertheless we decided to include this datasets for several reasons: 1) Even with missing depth the dataset can contain precious information. 2) Although the exact depth is unknown, a estimation, e.g. 50 to 100 m or 100 to 200 m, is normally possible. 3) In future literature review we will find maybe additional information and we will be able to complete the depth. In this case it would be a pity to have excluded the database in advance, only due to a missing depth. From our experience, it was often possible to merge datasets from different literature sources. |
| 10, 9 | The authors state that no time series of measurements are considered. Is there a reason for this? Secondly, how exactly are the measurements reduced? | We excluded time series measurements only for the statistical evaluation (not from the database!). In case multiple measurements were available from one location, only one measurement was included in the statistical evaluation; the remainders were not taken into account. |
| 10, 10 | Here, likely again due to my limited geo-hydrological understanding, it is not clear why a dataset with an electrical balance error larger than 5% is deleted (and not i.e. flagged). Does this imply that such a measurement has to be wrong, or is it just imprecise? | Well, in the database is maintained and flagged. However, for the statistical evaluation these analyses were not considered, because a error greater than 5 % is by convention not precisely enough. |
| 10, 11 | After conditioning, only roughly half of the datasets were kept in the database. This seems like a lot of cleanup to me. It would be very helpful to understand, which of the three criteria given on p.9 L.32-34 ‚removed' most of the datasets. Were most datasets simply reduced (criteria 2.) or erroneous (1, 3)? Finally, if a considerable amount of the datasets have been deleted due to the 1st and 3rd criteria, did this affect mostly older datasets? | Again, the conditioning was done for the statistical evaluation only; no data was deleted from the (original) database of Hesse. Here are the rough proportions of the individual criteria on the conditioning: criteria 1 removed appr. 30%; criteria 2 removed appr. 60%, criteria 3 removed appr. 10%. And yes, criteria 1 and 3 most often deleted old/older datasets. |

| Fig. 1 | I am having some difficulties making use of this Figure. The sampling points are hard to identify and I almost overlooked them on first sight. I would suggest taking some contrast out of the figure, by using less-saturated fill colors for the geological units. Finally, I think it is enough to have Frankfurt marked on the map for orientation purposes. I found the amount of cities and the fact that they are abbreviated rather confusing. | Now, the caption includes a note regarding color and shape of the sampling points. The borders of the hydrogeological regions (HYR) are expanded and shown in dashed lines to distinguish them better from the geological background. We deleted roughly half of the cities and wrote out the remaining. We decided to keep some cities because most of them are mentioned in section 1.2. For orientation it might be helpful to keep these cities on the map. |
|---|---|---|
| Fig. 3-5 | I think an axis label can make these figures even clearer | Done. |
| Fig. 3 | Just out of curiosity: Do the authors have any possible explanation for the ,cold' outlier? | This is a well-founded question. Uldluft (1969) and Carlé (1975) cite a analysis of 1905 with a temperature of 41.5 °C, which fits well to the normal geothermal gradient and temperatures observed in the neighboring well (Solebohrung III, #674 and #675 in the data base). After a thorough review of the original literature, we came to the conclusion that the temperature of 14.9 °C given by Pickel (1975) and Käß and Käß (2008) in the analysis of 1909 is an error due to transposed digits. We modified it to 41.9 °C in the manuscript and flagged the value in the data base. |

**Database**

| Some of the cells in the Excel file have light grey or dark grey background. Although this is explained at the end of the dataset description, it would be convenient to have the explanation in the Excel file as well. Maybe as a comment, legend or extra page. | We inserted a new sheet "Info" were we inform the color code for all rows and columns. |
|---|---|
| The first row is empty and can be removed | Done. |
| From l. 1041, the format of the table changes considerably. This makes it really complicated to read the file automatically (i.e. with R or Python). | Yes, the database of Hesse ends in line 1040 and in the following lines are additional information and analyses for comparison. Unfortunately, it is not possible to provide the same format due to the data structure. We followed Your suggestion below and created separate sheets. |
| L.1046 – 1064: Here I have some difficulties understanding the presented data. It seems like this section should be a sheet of its own. The IDs start over with one again, which leads to duplicated IDs, which is not compatible with a relational data model. The same applies to the section from L.1066 to 1102. | See comment above. |
| The column 'Water containment system' (column L) defines the value space as '(well, spring)', but I can also find 'Borehole', which should be added. The values are capitalized, but the definition of the value space in the header is lowercase. It would be nice to keep this consistent. | There are some other entries like "shallow well", "adit", "hand drilling", "drainage". Thus we deleted the examples and added sealing. |

| | |
|---|---|
| The column 'Static water level' has 'meter' as a unit, but I can find strings in the values (i.e. 'Artesian' or 'Free overrun'). I would suggest keeping this column atomic and of single data type. If the static water level cannot be expressed in the specified unit, a possible path is to split it up into two columns. | Entries like "free overrun" or "artesian" indicate that the static water level is unknown as it is above the altitude of the sealing or the ground level. In these cases we could equalise the static water level with the altitude to avoid text entries in favour of digits. In doing so we would lose information, moreover such an approach is wrong from a hydrogeological perspective. Thus, we prefer to maintain the column like it is. |
| The column 'Rock type' (column R) contains question marks. Most other columns are kept empty if the information is not available. I would suggest either to keep this consistent or add an explanation of the difference if there is any. | All empty fields are filled with a question mark now. |
| Columns U,V,W,X: these columns are just labeled 'T.M,B,S'. For me, it is not clear what these columns specify. Maybe short information or a header can be added. | The title in lines 3 and 4 is modified and contains now: T = thermal [water], M = mineral [water], B = brine, A = acidulous We modified this abbreviation from "S" to "A". |
| Columns 'Cations' and 'Anions' (columns Y and Z) contain some broken formulas. I would suggest checking this formula, and remove the formula from cells that are intentionally empty | Corrected, fields with broken formulas are empty now. |
| The format of the date-times in column AB (Date of analysis) is not consistent. | Normally the date format is DD/MM/YYYY like defined in the title. Uncompleted dates are expressed with "x", like xx/xx/1905. Sadly, some authors do not inform dates of analyses. In this cases we wrote "before YYYY" (YYYY is here the publication date). For the sake of information we believe that this approach is better than a strict uniform data structure. |
| The column 'Temperature' (AG and AL) contains the string 'n.d.' and empty cells. I assume 'n.d.' is the abbreviation of 'not defined'. How can the temperature be not defined? It can be not observed, like in the empty cells, but what exactly does 'n.d.' mean in this context? | Column AF (former AG) informs the water temperature. There are no empty fields within this column, but always a number or "n.d.". We agree that the abbreviation n.d. (yes, we tough of "not defined") is misleading and replaced it here as well as in columns L (former M) and CN (former CO) by n.s. ("not specified"). These columns are fundamental and we preferred to keep the information "n.s." instead of an empty field in order to inform the reader that the information was not available for us. Otherwise you could think that we forgot to insert values or so. In this context we also changes figure 2. Column AK (former AL) informs the water temperature of the density measurement. There are only entries, where the density is informed as well. Otherwise fields remain empty. |
| The columns with chemical parameters (AV – DT) are all defined to be of numeric data types, but occasionally contain strings (i.e. 'Traces' or '<0.5'). What is the difference between 'Traces' and '< xx'? I personally would interpret i.e. ' < 0.5' (cell BK 350) as 'below the detection limit of Nitrate'. But in BK 344 (same column) I also find 'Traces', which I personally would interpret in the same way. In BK 352 I find the number 0.22, which is also '<0.5'. I understand, that it is | You are right, "< xx" means below the detection limit of xx. The problem is that especially in old analyses information on the detection limit are missing. Authors used to write "traces" or something comparable to inform the reader that they could somehow detect the ion or element quantitatively but not qualitatively. We believe that this information is helpful and important why we would like to keep text, although there are numeric columns actually. From our perspective it |

| | |
|---|---|
| challenging to harmonize this information from different data sources, nevertheless, I think it is important to make this great dataset more readily usable. A user will have to harmonize this anyway if he wants to load the database into a GIS or using a programming language. | is not possible to "translate" the information "traces" into a detection limit. |
| From column EL to GE, there are a lot of broken formulas. | Sorry, an ugly mistake. It is corrected. |
| First, I want to acknowledge that the authors added an ER diagram of the data model to the dataset, that's really helpful. However, I am struggling with some details:
▪ I think the relations are not really clear. What do they connect and what do they describe? Following the relations, a record of 'Physical-chemical parameters' relates a 'Metadata' by the fields 'Temperature' to 'Thermal water'. First, I can't find 'Thermal water' in the database and second, I really have some difficulties imagining this relationship. Is this relation a foreign key, like usually noted for database ER diagrams? How can the actual temperature be used for referencing entities? I feel like I don't really get the idea here.
▪ In the Metadata entity, the 'Coordinates GK H' self-references the UTM Northing. This is obviously not a self-reference in a technical sense, as the Gauß-Krüger Hochwert will never match the UTM Northing numerically. Additionally, whatever this relation is describing, does not apply to the GK Rechtswert and UTM Easting, as there is no relation indicated. The Easting is rather self-referencing the Altitude. Without further explanation, I cannot really make sense of this information.
▪ I think I am completely on the wrong track interpreting the ER diagram, here. As a database engineer, I use ER diagrams a lot for database design and I might have a very narrow view on them. I guess there is only missing a short textbox explaining the meaning of an entity and a relation here and all confusion is cleared up. Maybe a legend is already enough. | We guess the reason why You are struggling with our ER diagram is that as database engineer You are the professional and we have no or very limited experience with ER diagrams. For us, it is very interesting to read Your feedback!
Well, we seize Your suggestion in the last bullet point and added a sentence in the caption of the ER diagram which might solve the confusion: "Black arrows mean that entries in fields at the end of the arrow depend on entries in the fields at the beginning of the arrow by formulas." |